# *Nypa fruticans* wurmb (Mangrove Palm) Fruit Kernel: Effect on physicochemical, sensory evaluation and shelf-life of wheat biscuits

Md. Ripaj Uddin[1]*, Md. Salim Khan[1], Md. Khairul Islam[2], Muhammad Abdullah Al Mansur[1], Md. Selim Reza[3], Md Hamedul Islam[4], Mehedi Hasan[1], Sharmin Ahmed[1], Sirajum Monira[5], Abubakr M. idris[6,7]

1 Institute of National Analytical Research and Service (INARS), BCSIR, Dhaka, Bangladesh, 2 Institute of Mining, Mineralogy and Metallurgy (IMMM), Bangladesh Council of Scientific and Industrial Research (BCSIR), Joypurhat, Bangladesh, 3 Department of Biochemistry & Molecular Biology, Jahangirnagar University, Savar, Dhaka, Bangladesh, 4 Departrnent of Pharmacy, Dhaka International University, Dhaka, Bangladesh, 5 Geological Survey of Bangladesh, Ministry of power, energy & mineral resources, Dhaka, Bangladesh, 6 Department of Chemistry, King Khalid University, College of Science, Abha, Saudi Arabia, 7 Research Center for Advanced Materials Science (RCAMS), King Khalid University, Abha, Saudi Arabia

* md.ripajuddin@gail.com

## Abstract

The global biscuit market is shifting towards healthier, gluten-free options, demanding novel and sustainable ingredients. This research introduces Nipa palm kernel flour (NFK) an underutilized, gluten-free resource from the mangrove *Nypa fruticans* Wurmb as a promising functional ingredient for biscuit production. Mature fruits were collected from the Sundarbans, Bangladesh, to develop nutritious snacks and valorize this sustainable resource from an underutilized plant. Kernels were extracted, blanched, dried, and milled into flour. Biscuits were formulated by substituting refined wheat flour with 0% (control), 10%, 20%, and 30% NFK. The physical, nutritional, sensory, and shelf-life properties of the biscuits were comprehensively analyzed using standard methods. NFK incorporation significantly improved the biscuit's nutritional profile. The 30% NFK biscuit exhibited a 38% reduction in fat, a seven-fold increase in crude fibre (from 0.19% to 1.35%), and a substantial boost in essential minerals, with iron content more than doubling to 53.3 mg/kg. Incorporating NFK significantly altered the biscuits' color profile, reducing lightness (L: 65.30 to 55.40) and yellowness (b: 25.80 to ~22.50–23.10) while increasing redness (a*: 4.50 to ~5.85–6.10), resulting in a desirable, darker red-brown hue. Crucially, all NFK-enriched biscuits received high sensory acceptability scores (≥7.85 on a 9-point scale), with no significant difference in overall liking compared to the control. Microbiological analysis confirmed product safety and a shelf life of over 12 months. The incorporation of *Nypa fruticans* kernel (NFK) flour at 15% substitution significantly enhanced the biscuits' functional profile, increasing total phenolic content by 212% (to 265.8 mg GAE/100g) and flavonoid content by 174% (to 89.1 mg QE/100g). This

**Data availability statement:** All data are available in the manuscript.

**Funding:** The Deanship of Research and Graduate Studies at King Khalid University for funding this work through the Small Research -Group under grant number RGP.1/266/46.

**Competing interests:** The authors have declared that no competing interests exist.

directly boosted antioxidant capacity, with DPPH $IC_{50}$ decreasing 3.3-fold (to 3.8 mg/mL) and FRAP value increasing 4.6-fold (to 8.2 mmol $Fe^{2+}$/100g), transforming the biscuit into a potent bioactive food carrier. NFK is a promising gluten-free biscuit ingredient. Future research should confirm large-scale production, in-vivo bioavailability, and precise compound characterization, supporting sustainable mangrove use and healthy food choices.

## 1. Introduction

Biscuits are well known baked foods with low moisture content and have long shelf life, which are considered as a convenient, inexpensive snack among all the age groups across the world [1]. With the increased consumption of biscuit products, there is also an increasing need for their dietary functionality [2]. Biscuits contain various ingredients such as flour, sugar, milk, eggs, fats or shortening and flavoring ingredients [1]. The flour-based component of biscuits used as raw material and largely are wheat based and have a high gluten content is (approximately 12–14%) [3]. The application of fat replacers, such as roasted and germinated chickpea flours, significantly alters the techno-functional and microstructural properties of biscuits [4], while the substitution of shortening with extra virgin olive oil notably enhances the nutritional profile and modifies the sensory characteristics of traditional Italian Cantuccini [5]. Nonetheless, due to increasing consumer preference for natural, healthy and functional foods and a high level of competition in the market, food companies are beginning to replace traditional flours with new ones including corn, flax seed, potato flour and defatted rice bran or other non-gluten grains like beans in developing formulations [6]. The Nipa palm (*Nypa fruticans* Wurmb) is a mangrove palm, and the only member of the Genus: Nypa. Known as "golpata" in Bangladesh, this plant is common throughout the mangrove forests of Asia, Oceania and east Africa [7,8].

In Bangladesh, it exists in the coastal and tidal area especially in the Sundarbans [9]. Although the plant, *N. fruticans* is available commercially in the wild, rural farmers residing in the southwestern districts of Patuakhali, Bagerhat, Khulna and Satkhira grow this plant as a cash crop in agricultural fields for domestic purpose. Its tough leaves are an important commodity, often employed for thatching houses in these areas [10]. Besides its construction uses, the inflorescences of *N. fruticans* can be tapped to collect a sap that may be fermented to vinegar and alcohol or boiled for brown sugar [11]. In addition, *N. fruticans* is highly productive in fruit production and can bear fruits for several times a year. Fruit bundles have 30–50 individual fruits, and each ripe fruit consists of a brownish epicarp, fibrous mesocarp, thin endocarp and central whitely developed kernel [12]. This palm kernel, freshly "white" from mature fruit of *N. fruticans* is potential to be converted to flour.

*N. fruticans* fruit kernel meal is a rich ingredient in terms of nutritional composition, presenting high carbohydrates (74.57 to 87.56%) and dietary fibers (0.18–17.68%), but low amounts of lipids (0,40–12,25%) and proteins (2.66–8.17%) [13,14]. The kernel is also a good source of vitamins A (2.57 mg/100 g) and C (20.36 mg/ 100 g)

as well as essential minerals such as magnesium, potassium, calcium, sodium, manganese iron and copper. Despite the existence of anti-nutritive factors such as hydrogen cyanide, tannins, phytic acid and oxalates in the ripe fruit kernels of *N. fruticans* cultivars have concentrations that do not exceed lethal poisoning for humans (2.5 g/100 g) [13]. Consequently, flour of *N. fruticans* fruit kernel may play a role as a nutritive factor in food products. In the context of biscuit sector competition and market pressure to gluten-free product, *N. fruticans* fruit kernel flour can be considered as a potential substitute for wheat flour [15]. Introduction of this ruin-free flour into biscuit manufacturing would not only address the requirement of health-oriented end-consumers with special dietary needs but using waste nutrition product it will also lead to sustainability and promote new product development [16]. The description of NFK valorization still can add to national economic development and infrastructure integrity and food security as well as job opportunities.

However, a significant knowledge gap exists regarding the application of underutilized mangrove resources like NFK in gluten-free, nutrient-fortified baking. While conventional wheat biscuits are often criticized for their high glycemic index, low dietary fiber, and micronutrient density, and reliance on extensive agricultural systems [17], research into alternative flours has predominantly focused on common legumes and tubers. Previous studies on composite biscuits have incorporated ingredients like coconut flour and defatted peanut cake [18], yet the unique functional and nutritional properties of NFK remain largely unexplored in the literature. This study directly addresses this gap by systematically investigating how NFK flour inclusion specifically addresses the problems associated with standard wheat biscuits by enhancing their protein, mineral, and dietary fiber content, thereby proposing a novel, sustainable ingredient that leverages coastal biodiversity to improve nutritional outcomes.

The objective of the present study is to investigate the nutritional composition and consumer acceptability of the biscuits with N-kind flour from *Nypa fruticans*, schard kernel and further to assess changes in their nutrient content after baking. By producing biscuits with NFK as functional ingredient, a high-quality gluten-free and healthy snack item can be introduced while enhancing the sustainable utilization of an undervalued resource.

## 2. Methods and materials

### 2.1. Source of *Nypa fruticans* fruit and Ethical Approval

The mature fruits of *Nypa fruticans* were collected from the estuarine mangroves of the Sundarbans in Bangladesh (about 21.5ºN - 22.5ºN, 89.0ºE – 90.0ºE) during the fruiting season period (02/01/2024–30/03/2024). Fruits were chosen for the uniform brown colored exocarps, comparable size and not involving physically damaged or microbial infested. A voucher specimen was identified (Voucher Specimen No: DACB 12345). It fits within the context of the previous studies which have been undertaken on underutilised mangrove resources [19,20]. This research was approved ethically by the Committee of Clinical Pharmacy & Pharmacology, Faculty of Pharmacy, Dhaka International University (Ref No: CPP/DIU/EC/22). Fresh fruits of *Nypa fruticans* were collected from the Sundarbans Mangrove Forest or specific river estuary in Bangladesh. No specific permits were required from Forest Department of Bangladesh for the collection of *Nypa fruticans* fruits at this site, as the area is publicly accessible and the species is not protected or endangered under national or local regulations in this region. The collection was for academic research purposes and did not involve disturbance of protected land or threatened wildlife.

### 2.2. Kernel extraction and flour preparation

NFK flour Kernels (Fig 1) from *Nypa fruticans* (NF) were milled according to the modified method for palm kernels [21,22]. The fruits are optimally harvested at 90–100 days after blossoming, when the endosperm has solidified to a firm, white consistency and the epicarp readily detaches. The fibrous epicarp was removed manually by steel knife and the nuts were cracked by opening it to remove the kernel. The kernels were washed with drinking water before rinsing twice with 0.5% solution of sodium metabisulfite to avoid enzymatic browning. Thereafter, samples were blanched at 80 °C for 5 min to inactivate polyphenol oxidase and lipase enzymes [23]. The blanched kernels were oven dried at 60 °C for 48 hours

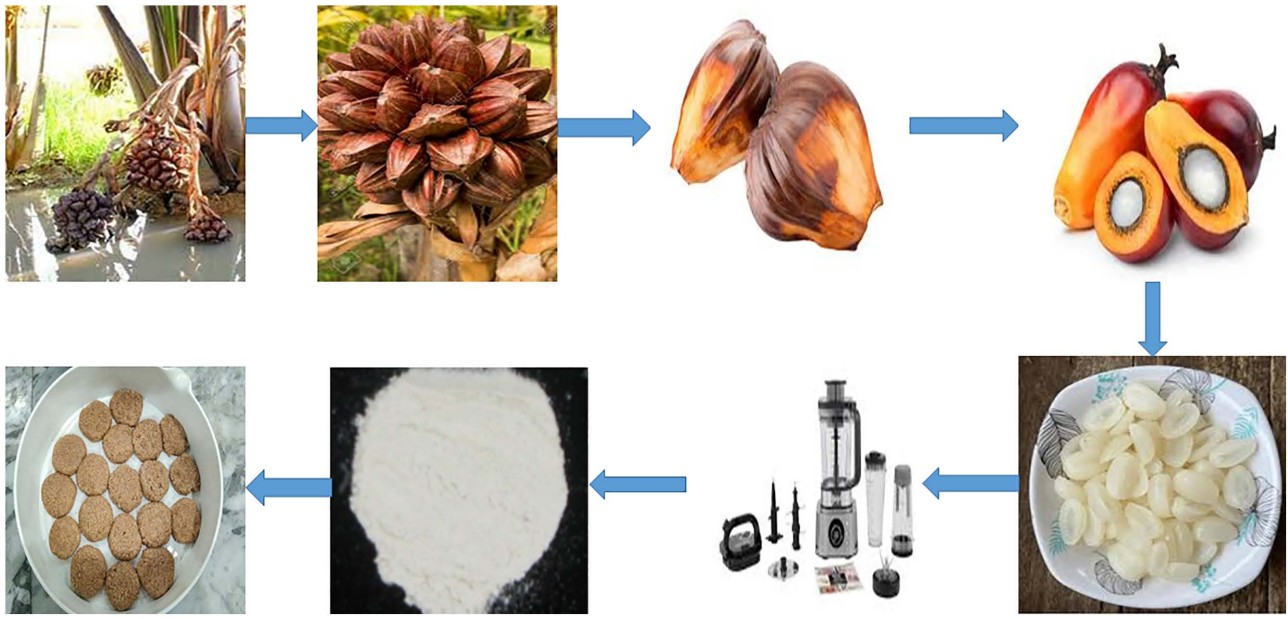

**Fig 1. Making biscuits utilizing NFK Flour.**

to retain heat-labile nutrients [24] and ground with hammer mill (Retsch SK 100). The dried minced barley was passed through an 80-mesh screen (180 μm) in order to ensure that the particle size was consistent with that of white wheat flour [25]. Finally, the NFK flour was sealed in nitrogen-flushed high-density polyethylene (HDPE) bags with oxygen absorbers and kept in a desiccator 25±2 °C to avoid oxidative deterioration until ready for analysis [26].

### 2.3. Proximate composition analysis of NFK flour

The proximate composition of the samples including moisture, ash, crude protein, crude fat, crude fiber, and total carbohydrates was determined using established official methods. Moisture content was determined gravimetrically according to AOAC 934.01. Briefly, a known weight of the sample was dried to a constant weight in a hot-air oven (Memmert GbH+Co. KG, Schwabach, Germany) at 105 °C. The moisture content was calculated as the percentage of weight loss due to water evaporation using the equation No. 1 [27]:

$$\text{Moisture (\%)} = \frac{W - W_1}{W} \times 100$$

(1)

Where, W=Weight of fresh sample (g), and W1=Weight of dried sample (g).

Ash (total mineral matter) was analyzed via the direct method outlined in AOAC 942.05. The gross energy value was subsequently calculated from these results. A pre-weighed sample was incinerated in a muffle furnace (Nabertherm GbH, Lilienthal, Germany) at 550 °C for 5 hours until a constant white ash was obtained. The ash content was expressed as a percentage of the original sample weight calculated as equation No. 2 [27]:

$$\text{Ash (\%)} = \frac{W_1}{W} \times 100$$

(2)

Where, W=Weight of sample (g) and W1=Weight of ash (g).

 

Crude protein content was estimated by determining the total nitrogen content using the Kjeldahl method as per AOAC 984.13 (A-D). The nitrogen content was converted to crude protein using a specific conversion factor as shown in the equation No. 3 [27,28]:

$$\text{Crude Protein (\%)} = \frac{(B - S) \times N \times 1.007}{W} \times F$$

(3)

where, B = mL of acid blank, S = mL of acid sample, N = Normality of acid, W = Weight of sample (g) and F = 5.70 (nitrogen-to-protein conversion factor) as recommended by the codex standards.

Crude fat content was determined by continuous solvent extraction using a Soxhlet apparatus following the AOAC 920.39 (A) method for crude fat. A known weight of the dried sample was subjected to extraction with petroleum ether (boiling point range 40–60 °C) for 6 hours as a percentage equation No. 4 [28]:

$$\text{Crude Fat (\%)} = \frac{W_1}{W} \times 100$$

(4)

Where, W1 = Weight of fat (g) and W = Weight of sample (g)

Crude fiber content was analyzed according to the AOAC 962.09 method. The defatted sample was subjected to sequential digestion with 1.25% $H_2SO_4$ and 1.25% NaOH solutions under controlled conditions to remove soluble carbohydrates and proteins. The remaining insoluble organic residue was filtered, dried, and weighed, then ashed in a muffle furnace at 550 °C. The loss in weight upon ashing represents the crude fiber, calculated as equation No. 5 [28]:

$$\text{Crude Fiber (\%)} = \frac{W_1 - W_2}{W} \times 100$$

(5)

Where, W1 = Weight of residue after digestion (g), W2 = Weight of ash (g) and W = Weight of sample (g)

The percentage of total carbohydrates was obtained by subtracting the sum of the percentages of moisture, ash, crude protein, and crude fat from 100 equation No. 6 [29,30]:

$$\text{Total Carbohydrates (\%)} = 100 - (X + Y + A + B)$$

(6)

Where, X = Moisture %, Y = Ash %, A = Crude Protein % and B = Crude Fat %.

Finally, the energy content was calculated from the proximate composition using the modified Atwater general factors as described by Merrill and Watt which assign 4 kcal/g for proteins, 9 kcal/g for fats, and 4 kcal/g for carbohydrates. The gross energy was calculated using the formula No 7 [29,31]:

$$\text{Energy (kcal/100g)} = 4 \times A + 9 \times B + 4 \times C$$

(7)

While, A = Crude Protein (g), B = Crude Fat (g) and C = Total Carbohydrates (g).

## 2.4. Mineral and toxic metal analysis with quality control

The mineral and toxic metal profile of the NFK powder was determined using atomic absorption spectrophotometry (AAS) following a rigorous digestion and analytical protocol to ensure accuracy and precision. All procedures adhered to standard methods from the American Public Health Association (APHA) [32,33] and the quality assurance system of ISO/IEC 17025:2017 [34]. All glassware and volumetric equipment were meticulously cleaned by soaking overnight in a 2% (v/v) nitric acid ($HNO_3$) solution prepared with deionized water (18.2 MΩ·cm resistivity, 0.2 µS/cm conductivity) and thoroughly rinsed prior to use to prevent any cross-contamination [35–37].

A precisely weighed 2-gram sample of the homogenized kernel powder was subjected to a wet digestion process in a calibrated fume hood. The sample was placed in a high-quality borosilicate glass beaker and digested with 20 mL of 70% analytical grade $HNO_3$ (Siga-Aldrich, Germany) on a hot plate at 90 °C for approximately two hours until a clear digestate was obtained. The digested sample was then cooled, filtered through a Whatman No. 42 ashless filter paper to remove any particulate matter, and the filtrate was quantitatively transferred into a 100mL volumetric flask where the volume was made up to the mark with deionized water. For the determination of toxic metals at very low concentrations, a more sensitive graphite furnace AAS (GF-AAS, Varian AA240Z with GTA 120) was employed for the analysis of Cr, Cd, As, and Pb. The analysis of essential minerals (Fe, Mn, Cu, Zn, Mg, Ca) was conducted using flame atomic absorption spectroscopy (FAAS; Varian SpectraAA 240FS/220). Concentrations of the major cations (Na and K) were determined by a calibrated flame photometer (Jenway PFP7, UK).

The entire analytical procedure followed a rigorous Quality Assurance and Quality Control (QA/QC) plan. The method was validated and calibrations were carried out using certified reference materials (CRMs) and high-purity reagents (Scharlau, Siga-Aldrich). Standards Stock standard solutions (1000 mg/L) of each element were appropriately diluted to give calibration curves. Procedure reliability was ascertained by spiking experiments, duplicate analyses of samples, independent standard checks and the analysis of procedural blanks in each batch. Data quality and precision were controlled through triplicate analysis of a NIST-traceable CRM in combination with the use of a control chart (99% confidence limit) to monitor results. The method detection limit (MDL), calculated as three times the standard deviation of the blank signal, was determined to be 0.005 mg/L for Cu, 0.01 mg/L for Cr, 0.001 mg/L for Pb, 0.05 mg/L for Cd, 0.1 mg/L for Mn, and 0.07 mg/L for Zn. The final concentration of each element in the sample was calculated after blank subtraction using the formula No 8:

$$\text{Element Concentration (mg/kg)} = \frac{(C \times V \times D)}{M} \tag{8}$$

where C is the concentration of the element in the digested solution (mg/L) obtained from the calibration curve, V is the final volume of the digest (L), D is any additional dilution factor, and M is the mass of the original sample (kg).

## 2.5. Extraction and phytochemical analysis of composite flour

The composite flours (wheat flour blended with NFK powder at 0%, 5%, 10%, and 15% substitution levels) were analyzed for key phytochemical properties. A methanolic extract was prepared for each composite flour sample. Briefly, 5 g of flour was mixed with 50 mL of 80% aqueous methanol and agitated on an orbital shaker at 150 rpm for 24 hours at room temperature. The mixture was then centrifuged at 4000 rpm for 15 minutes, and the supernatant was filtered through Whatman No. 1 filter paper. The residue was re-extracted twice. The combined supernatants were concentrated under reduced pressure at 40°C using a rotary evaporator. The dried extract was reconstituted in methanol to a known concentration (e.g., 10 mg/mL) for subsequent analyses [38].

**2.5.1.  Determination of Total Phenolic Content (TPC).**  The TPC of the extracts was determined using the Folin-Ciocalteu method [38]. Briefly, 0.5 mL of appropriately diluted extract was mixed with 2.5 mL of 10% Folin-Ciocalteu reagent and allowed to stand for 5 minutes. Then, 2 mL of 7.5% sodium carbonate solution was added. The mixture was incubated in the dark for 60 minutes at room temperature. The absorbance was measured at 765 nm using a UV-Vis spectrophotometer. Gallic acid was used as a standard, and the results were expressed as milligrams of Gallic Acid Equivalents (GAE) per 100 grams of flour (mg GAE/100 g).

**2.5.2.  Determination of Total Flavonoid Content (TFC).**  The TFC was estimated using the aluminium chloride colorimetric method [39]. An aliquot of 0.5 mL extract was mixed with 1.5 mL of methanol, 0.1 mL of 10% aluminium chloride, 0.1 mL of 1 M potassium acetate, and 2.8 mL of distilled water. The mixture was vortexed and allowed to stand at

room temperature for 30 minutes. The absorbance was measured at 415 nm. Quercetin was used as a standard, and the results were expressed as milligrams of Quercetin Equivalents (QE) per 100 grams of flour (mg QE/100 g).

### 2.5.3. Determination of DPPH Radical Scavenging and Ferric Reducing Antioxidant Power (FRAP) Assay.

The free radical scavenging activity was determined using the 2,2-diphenyl-1-picrylhydrazyl (DPPH) method [40]. Briefly, 2 mL of 0.1 mM DPPH methanolic solution was added to 2 mL of various concentrations of the extract. The mixture was shaken vigorously and incubated in the dark for 30 minutes. The decrease in absorbance was measured at 517 nm. A methanol-only solution served as the control. The percentage inhibition was calculated, and the $IC_{50}$ value (concentration required to scavenge 50% of DPPH radicals) was determined from a calibration curve using Trolox as a standard.

The reducing power was assessed using the FRAP assay [41]. The FRAP reagent was prepared fresh by mixing 300 mM acetate buffer (pH 3.6), 10 mM TPTZ (2,4,6-tripyridyl-s-triazine) in 40 mM HCl, and 20 mM $FeCl_3 \cdot 6H_2O$ in a 10:1:1 ratio. An aliquot of 100 μL extract was added to 3 mL of the FRAP reagent and incubated at 37°C for 10 minutes. The increase in absorbance was measured at 593 nm. Ferrous sulfate ($FeSO_4 \cdot 7H_2O$) was used to prepare a standard curve, and results were expressed as millimoles of Ferrous Equivalents ($Fe^{2+}Eq$) per 100 grams of flour (mmol $Fe^{2+}Eq/100 g$).

## 2.6. Biscuit formulation and preparation

Biscuits were formulated by partially substituting wheat flour with NFK powder to create nutrient-rich, low-fat products. A control biscuit (0% substitution) and several experimental biscuits with varying levels of NFK substitution (e.g., 10%, 20%, 30%) were developed based on a standard creamery-style short dough biscuit formula, adapted from methods commonly used in cereal science research [42,43] (Manley, 2011; Pareyt & Delcour, 2008). The control formulation consisted of wheat flour (1000 g), powdered sugar (250 g), fresh whole eggs (4 units, approximately 200 g), sodium chloride (5.0 g), sodium bicarbonate (2.0 g) as a leavening agent, dried yeast (4.0 g) for additional aeration, a reduced amount of shortening (30 g) to align with the low-fat objective, and vanilla essence (6 g) for flavour. The added water for dough formation was standardized to 500 ± 20 Brabender Units (BU, arbitrary units) across all batches and assessed empirically. The dough, which was creamed in order to aerate it uniformly and develop the crumb structure, consisted of the following constituents:

Briefly, shortening and sugar were creamed in a planetary mixer (KitchenAid 5K5SS, USA) with a flat beater at medium speed for 5 min. Then the eggs and vanilla extract were added gradually with continued mixing until it became a pale, airy emulsion. In a separate bowl, premixed wholewheatflour, NFK powder if used in the recipe, soda and yeast ascorbic acid were made up. This dry blend was added to the creamed portion and was combined at low speed minimum mixing to avoid overdevelopment of gluten structure. Water was added gradually until a stiff, muddy dough formed. Finally, the dough was rested for 10 min at 25 °C to achieve complete hydration and gluten relaxation. The relaxed dough was sheeted to 6.0 mm thickness by means of a small scale sheeter (Rondo GbH, Switzerland) and cut into round sheets of 50 cm in diameter. The biscuits were transferred to a greased aluminium baking tray and baked in pre-heated forced convection oven (Model UNE400, Memmert GbH + Co. KG, Germany) at 180 °C for 12–15 minutes until light golden brown colour was obtained. The biscuits were then cooled at room temperature on wire racks for 1 h (to ensure that moisture had a chance to redistribute and the structure to develop).

The cooled biscuits were packaged in metallized polyester pouches with high oxygen and moisture barrier properties and heat-sealed. They were then stored at 20 °C in a desiccator containing silica gel to retard moisture uptake and maintain stability until analysis. Baking loss was determined gravimetrically as the percentage of weight lost during baking due to water evaporation and the loss of volatile components, calculated using Equation (9) [44–46]:

$$\text{Baking Loss } (\%) = \frac{W_1 - W_2}{W} \times 100$$

(9)

Where W is Weight of dough before baking (g), $W_1$ is Weight of dough before baking (g) and $W_2$ is Weight of biscuit after baking (g).

## 2.7. Analysis of physical and chemical properties of biscuits

Physical properties of the biscuits produced were investigated after 24 hours at room temperature and controlled humidity. The diameter and the thickness of ten randomly taken biscuits from each batch were measured using a digital vernier caliper (Mitutoyo, Corp., Japan) at the accuracy of 0.01 mm. Weight of individual biscuits was determined using analytical balance, and average weight for each formulation was calculated. The spread ratio, an important quality parameter characterized as the width of crackers per thickness was measured following AACC International Method 10–50.05 [44]. This parameter is the primary scale for baking performance and end-product quality [47].

$$\text{Spread Ratio} = \frac{\text{Width}}{\text{Thickness}} \tag{10}$$

Texture profile of biscuit (in term of hardness represented by the peak force) was determined using Texture Profile Analysis (TPA). A calibrated texture analyzer (TA. XTplus, Stable micro system Ltd, UK) fitted with a three-point bending rig (HDP/3PB) and 50 kg load cell was employed [48]. Each biscuit was supported on 2 parallel bars separated by 40 mm and broken with a downwardly moving blade. The probe was moved at a pretest and post speeds of 1.0 mm/s, a test speed of 3.0 mm/s and a trigger force of 5 g. The highest force (N) recorded during penetration of the probe represented the hardness value. At least ten replicates of each batch were analyzed. The surface colour on the top and bottom surfaces of the biscuits was measured by using a HunterLab colorimeter (ColorFlex EZ, Hunter Associates Laboratory, Inc., USA), calibrated against white and black standard tiles.

The CIE L, a, b* colour space values were recorded where L* represents lightness (0 = black, 100 = white), a* represents the green-red component (negative values indicate green, positive values indicate red), and b* represents the blue-yellow component (negative values indicate blue, positive values indicate yellow). The total colour difference (ΔE) between the control and NFK-substituted biscuits was calculated using the equation 11 [49,50]:

$$\Delta E = \sqrt{[(L - L_0)^2 + (a - a_0)^2 + (b^* - b_0)^2]} \tag{11}$$

where $L_0$, $a_0$, and $b_0$ are the colour values of the control biscuit. Furthermore, the chemical composition of the final biscuit products, including moisture, ash, crude protein, crude fat, crude fiber, and total carbohydrate content, was determined using the same official AOAC [27].

pH were determined using pH meters (Model: HannaHI-255, USA). The concentrations of anions such as $Cl^-$, and $SO_4^{2-}$, and were analyzed by ion chromatography (Model: HIC-10A super, Shimadzu, Japan), while $PO_4^{3-}$ was investigated using an UV–Vis spectrophotometer (model: UV-1650PC, Shimad Zu, Japan).

## 2.8. Sensory evaluation

The sensory acceptability of biscuit formulations with *Nypa fruticans* kernel (NFK) substitution was evaluated through an affective consumer acceptance test. A panel (n = 120) of untrained consumers, recruited based on their frequency of consuming baked goods and absence of relevant allergies, participated [51,52]. Sensory testing was conducted [10/01/2024–10/01/2025] in a dedicated laboratory with isolated booths, following International Standards (ISO, 2007) [53]. The environment was maintained under controlled white light and a constant ambient temperature of 22 ± 1 °C to minimize external influence. A complete block design was used, where all panelists evaluated all samples including the 0% NFK control and the experimental treatments in a monadic and randomized order. Quarter-biscuit pieces were laid out

on white, odorless plastic plates at room temperature and were coded with a three-digit code in order to avoid bias. Subjects rinsed their palate with distilled water between evaluations. The consumers scored the appearance, color, aroma, flavor, texture and aftertaste on a nine-point hedonic scale (1 = dislike extremely; 9 = like extremely) [54,55]. This approach effectively confirmed the strong positive effect of NFK incorporation on consumer liking and perception.

## 2.9. Microbiological analysis and shelf-life study

The microbiological safety and stability of the developed low-fat biscuits were followed during storage to establish shelf-life. Study of shelf-life Shelf-life study was performed by keeping the packed biscuit samples processed from each formulation (control and NFK-substituted) at room temperature conditions (25±2 °C, 65% relative humidity) for a period of six months. The microbiological analyses of the bread samples were conducted immediately after baking and cooling (time zero) and repeated at monthly intervals to monitor microorganism proliferation in accordance with the standard methods recommended by APHA, 2018 [32] and ICMSF, 2018 [56]. The examination included the determination of total viable microbial load, indices for hygiene and fecal contamination, as well as directed search for bacterial pathogens. For all the analyses, a 10 g sample of each homogenized biscuit was aseptically weighed and placed in a sterile stomacher bag containing 90 mL of buffered peptone water (BPW; Himedia, India) to have a 1:10 stock dilution. The homogenate was mixed in the stomacher (Model 400, Seward Ltd., UK) for 2 min to achieve a homogeneous suspension. Decimal dilutions ($10^{-1}$ to $10^{-5}$) of the first homogenate were prepared in sterile 0.1% peptone water for plating. The Total Viable Count (TVC) of heterotrophic bacteria, a primary parameter reflecting the overall microbiological quality and spoilage of fish, was enumerated using spread plating technique on Plate Count Agar (PCA). Briefly, 0.1 mL of proper dilution were spread on the surface of pre-poured PCA plates (three replicates) and incubated aerobically at 36°C for 48±2 h [57]. After incubation, colonies were counted and the amounts were calculated as colony forming units per gram of sample (CFU/g) according to formula No. 12 [58]:

$$\text{TVC (CFU/g)} = \frac{\text{Number of colonies} \times \text{Dilution Factor}}{\text{Volume of culture plated (mL)}}$$

(12)

Quantitative microbial risk assessment was carried out by using a three-tube Most Probable Number (MPN) method for coliforms, indicator organisms and others. The presumptive test was performed, inoculating 1 mL of the $10^{-1}$, $10^{-2}$, and $10^{-3}$ dilutions into a test tube with Lauryl Tryptose Broth (LTB) with Durham's tubes. The tubes were kept in a 37 °C incubator for total coliforms and the incubation temperature for fecal coliform was 44.5 °C. Tubes that demonstrated gas production and turbidity were regarded as positive. These strains were subsequently subcultured into Brilliant Green Bile Broth (BGBB) for a confirmed test, and incubated as above. The MPN value per 100 g was calculated using standard MPN tables for the number of positive tubes at each dilution.

Furthermore, tested for major foodborne pathogens in the samples statements of selective. For Escherichia coli, 0.1 mL of homogenate was plated on Eosin Methylene Blue (EMB) agar and incubated at 37 °C for 24 h, bacteria consisting in metallic green sheen colonies were considered as presumptive. For Salmonella and Shigella spp., selective enrichment was carried out and plated on *SS agar*, before being incubated at 37 °C for 24 h [59]. The characteristic colorless colonies (for *Salmonella*) and Black centered ones for *Shigella* were observed. *Vibrio spp.* the detection was carried on Thiosulfate-Citrate-Bile Salts-Sucrose (TCBS) agar, at 37 °C for 24 h, and yellow-green colonies indicated the fermentation of sucrose. The presence of these pathogens was a critical component for product safety in 25 g of sample.

## 2.10. Statistical analysis

All chemical and sensory values of biscuits prepared were statistically analyzed to compare the difference among biscuit formulations by a Completely Randomized Design (CRD) [60,61]. Results of all laboratory analyses (n = 3) are expressed

as mean ± standard deviation. Statistical significance was calculated with one-way analysis of variance (ANOVA) using OriginPro software [62] with p < 0.05. Mean separations were done by Duncan's Multiple Range Test (DMRT) for the significant factors [63]. Second, sensory data (n = 120 panelists) were analyzed with ANOVA, along with Tukey's HSD for multiple comparisons which is efficient in case of large and equal sample size [64]. The equation of the statistical model for the experiment becomes:

$$Y_{ij} = \mu + \tau_i + \varepsilon_{ij}$$ (13)

where Yij is the growth status of each child at baseline, indicating whether children presented with stunting or not.

where: $Y_{ij}$ is j-th observation of the i-th treatment, $\mu$ is overall mean, $\tau_i$ is treatment effecto fthe i-treatment and $\varepsilon_{ij}$ is the random error associated to j-th observation i- thtreatment assumed to be independentl and no rmally distributed with median given for zero and homogeneous variance.

## 3. Results

### 3.1. Physical properties of biscuits

The incorporation of NFK flour significantly altered key sensory attributes, which are important factors determining acceptance by consumers and quality of products (Table 1). Most pronounced was the alteration of dimension-related features. Biscuit width increased significantly with increasing level of substitution of NFK from 52.10 mm in the control to 55.15 mm in biscuit incorporated with 30% NFK. On the other hand, thickness decreased significantly but slowly from 8.20 mm to 7.55 mm.

The spread ratio increased from 6.35 in the control to 7.30 in the biscuit formulation containing 30% NFK flour. This increased spread is a desirable characteristic for biscuits, contributing to a more aesthetically pleasing appearance. The effect is likely due to the dilution of wheat gluten and the unique water absorption capacity and rheological properties of NFK fibre. These factors may reduce dough elasticity and increase extensibility during the early stages of baking, prior to crust formation, resulting in greater lateral flow. This result is consistent with those recently reported on date seed powder fortified biscuits [65] and may be due to the breakdown of gluten network and changes in dough rheology.

The textural attribute, hardness was highly affected by NFK addition; it decreased significantly from 35.20 N in the control to 27.90 N in the biscuit containing 30% NFK. This decrease in fracture force implies a softer and more brittle texture, which is commonly accepted by consumers. The softening may have been due to gluten dilution, as the formation of NFK particles can disrupt the protein network, interference of NFK particles in the starch-protein matrix and dietary fiber lubricity. This result is in agreement with a study on jackfruit seed flour [66], which also showed lower hardness values which is

**Table 1. Physical characteristics of biscuits made from NFK flour.**

| Parameter | Control (0% NFK) | 10% NFK | 20% NFK | 30% NFK |
|---|---|---|---|---|
| Width (W, mm) | 52.10 ± 0.45[c] | 53.25 ± 0.38[b] | 54.80 ± 0.52[a] | 55.15 ± 0.49[a] |
| Thickness (T, mm) | 8.20 ± 0.15[a] | 7.95 ± 0.12[ab] | 7.70 ± 0.18[b] | 7.55 ± 0.14[c] |
| Spread Ratio (W/T) | 6.35 ± 0.12[c] | 6.70 ± 0.09[b] | 7.12 ± 0.15[a] | 7.30 ± 0.11[a] |
| Weight (g) | 12.50 ± 0.25[a] | 12.45 ± 0.30[b] | 12.55 ± 0.22[c] | 12.48 ± 0.27[d] |
| Hardness (N) | 35.20 ± 1.50[a] | 32.80 ± 1.25[b] | 29.50 ± 1.40[c] | 27.90 ± 1.35[d] |
| Color (L*) | 65.30 ± 1.20[a] | 62.15 ± 1.05[b] | 58.90 ± 1.30[c] | 55.40 ± 1.15[d] |
| Color (a*) | 4.50 ± 0.25[c] | 5.20 ± 0.30[b] | 5.85 ± 0.22[a] | 6.10 ± 0.28[a] |
| Color (b*) | 25.80 ± 0.80[a] | 24.20 ± 0.75[b] | 23.10 ± 0.85[c] | 22.50 ± 0.70[c] |

All values are presented as mean ± SD (n = 3 for dimensions, weight, hardness; n = 10 for color). Means in the same row with different superscript letters are significantly different (p < 0.05) according to Duncan's Multiple Range Test.

indicative of the disruption of gluten-starch network due to increased dietary fiber and hence less compact structure. This softening interaction is desirable because it describes a typical technical problem with low-fat bakery products, i.e., keeping the tenderness and preventing a tough or firm texture.

The surface characteristics of the biscuits, as measured by CIE Lab, were appreciably influenced by NFK incorporation. The color analysis reveals a clear and significant trend as NFK substitution increases: biscuits become progressively darker, redder, and less yellow. Specifically, lightness (L) significantly decreases from 65.30 in the control to 55.40 at 30% NFK, indicating a marked darkening. Concurrently, redness (a) increases from 4.50 to a plateau between 5.85 and 6.10 at 20–30% NFK, enhancing the red-brown tones, while yellowness (b*) decreases from 25.80 to approximately 22.50–23.10 over the same range, reducing the golden hue. This change to a dark brown reddish hue is due to the natural color and chemistry of the endosperm of NFK flour. The higher amounts of fiber and sugar present in the MHS may have favored Maillard reaction and caramelization during baking at 180 °C, which could favour the formation of melanoidins as well as other coloured products. Such darkening effect has also been reported for fortified biscuits containing mango seed kernel powder [67]. It is also worth mentioning that this darker color was sensory favorable with high scores, so it must had been perceived as appealing and associated like a whole grain product full of nutrients. In contrast, biscuit weight was consistent among formulations, which suggest that NFK substitution did not affect the baking loss or dough handling. In summary, the inclusion of NFK flour improved the biscuits' physico-sensory properties favorabvly and produced a biscuit with higher spread ratio, softer texture and optimum dark brown colour acceptable to consumers which will cater for more healthful high quality bakery products presently demanded by consumers.

### 3.2. Proximate and mineral composition of NFK flour

The estimation (Table 2) of pH, salinity and mineral composition validates the chemical stability and fortification achieved by addition of NFK flour in biscuit formulations. Valuable minerals like calcium, iron were preserved fostering fortification and food safety. The near-neutral pH of NFK flour (6.4) was carried over into the final biscuit product (6.46), suggesting that although Maillard reaction and caramelization reactions occurred during baking, no significant acidic compounds were generated. This pH stability is also preferable for storage as it prevents the growth of numerous spoilage microorganisms-a similar pH range 6.3–6.5 was found on rice biscuits enriched with tamarind seed powder [68]. The observed decrease in salinity may be related to the recipe itself. The large amounts of low-salt ingredients (refined wheat flour and sugar) mixed with NFK flour caused a reduction in ionic species per unit mass, as reported for mixtures similar composite food systems [69].

The addition of NFK significantly enhanced the mineral content of biscuits. Importantly, concentrations of toxic minerals such Cd, Hg and As were not determined (LOQ: 0.01 mg/kg) in either flour or biscuits; however, the concentration Pb slightly increased from 0.76 to 0.85 mg/kg a.u. These values are far lower than the maximum allowed limits of Codex Alimentarius Commission (e.g., 0.1 mg/kg for Pb in cereals) and indicate the safety compliance of NFK to be used as food [30].

An abundant enrichment of key elements was obtained from this analysis. The content of Na went up from 125.09 to 241.43 mg/kg, being due to the leavening agent (sodium bicarbonate) and salt (NaCl) used in the formulation of rico bread. On the other hand, K levels slightly diminished from 3.75 to 3.50 mg/kg and may have resulted from its thermal degradation or leaching; otherwise K losses are likely also nutritionally irrelevant as a result of its low content in samples. The profile of bone-building minerals [70] in particular was improved with significant increases seen: Ca from 295.2 to 428.4 mg/kg and Mg from 360.3 to 384.1 mg/kg for example). Among these, Fe registering more than two-fold increase from 21.5 to 53.3 mg/kg followed by Mn was increased from 10.3 to 16.7 mg/kg were the highest increase. This high iron absorption is very beneficial, especially considering that many people around the world suffer from a lack of this element. The iron content of the NFK biscuit is significantly higher than that reported in biscuits fortified with quinoa (28–35 mg/kg) or amaranth (32–40 mg/kg) flour [71]. Zn and Cu levels also increased slightly, to 12.9 mg/kg and 2.94 mg/kg respectively.

**Table 2. Chemical Composition of Flour and Biscuits.**

| Parameters | Flour | Biscuits |
|---|---|---|
| pH | 6.40±0.05[a] | 6.46±0.05[a] |
| Na (mg/kg) | 125.1±6.3[b] | 241.4±12.1[a] |
| K (mg/kg) | 3.75±0.19[a] | 3.50±0.18[a] |
| Ca (mg/kg) | 295.2±14.8[b] | 428.4±21.4[a] |
| Mg (mg/kg) | 360.3±18.0[a] | 384.1±19.2[a] |
| Fe (mg/kg) | 21.5±1.1[b] | 53.3±2.7[a] |
| Mn (mg/kg) | 10.3±0.5[b] | 16.7±0.8[a] |
| Cd (mg/kg) | <0.01±0.005[a] | <0.01±0.005[a] |
| Cr (mg/kg) | 1.00±0.05[b] | 1.41±0.07[a] |
| Hg (mg/kg) | <0.01±0.005[a] | <0.01±0.005[a] |
| Cu (mg/kg) | 2.70±0.14[a] | 2.94±0.15[a] |
| Zn (mg/kg) | 10.1±0.5[b] | 12.9±0.6[a] |
| As (mg/kg) | <0.01±0.005[a] | <0.01±0.005[a] |
| Pb (mg/kg) | 0.76±0.04[a] | 0.85±0.04[a] |
| $SO_4^{2-}$ (mg/kg) | 9.8±0.5[b] | 45.3±2.3[a] |
| $PO_4^{3-}$ (mg/kg) | 70.6±3.5[a] | 63.2±3.2[b] |
| $Cl^-$ (mg/kg) | 8.17±0.41[a] | 3.66±0.18[b] |

*All values are presented as mean±SD (n=3). Means in the same row with different superscript letters are significantly different (p<0.05) according to Duncan's Multiple Range Test.*

The anions analysis showed that some changes were significantly related to the chemistry of recipe. The $SO_4^{2-}$ also showed a remarkable increment from 9.8 mg/kg to 45.25 mg/kg. This increase could be due to heat-induced decomposition of sulfur-containing amino acids such as those present in eggs in the presence of the leavening agent. On the other hand, $PO_4^{3-}$ decreased from 70.6 mg/kg to 63.2 mg/kg might be attributed that it is involved in chemical reaction or bonded with other minerals. A significant reduction in $Cl^-$ content from 8.17 mg/kg to 3.66 mg/kg was achieved. This is surprising in the presence of NaCl, and this decrease might be related to volatilization of chloride compounds or other interactions taking place during baking.

Biscuits made from NFK flour are successful formulations for the concentration of nutritionally important minerals specifically Fe, Ca, Mg and Zn in the end product. These results indicate that these biscuits are safe for consumption and can be used as dietary source for the essential elements [67]. The amount of mineral enrichment obtained with NFK is even higher than for the other novel fortifiers discussed in recent literature, further indicating its high potential as a functional ingredient to address micronutrient malnutrition [64].

### 3.3. Proximate composition of developed biscuits

Application of the raw NFK flour to produce baked biscuits included major physicochemical changes (Table 3), which is in line with increasing interest for sustainable functional food ingredients worldwide. The low moisture content in baked biscuit (2.91%) as compared to raw flour (7.1%) was due to loss of water during baking that lead to concentration of other components and gave good degree of microbial stability during storage). This figure is similar to that found in biscuits from other flours such as mango kernel (2.8–3.5%) and watermelon rind powder (3.1–3.8%) [44,67] and also within the desirable range for products with long shelf-life. It should be noted that when the ash content is reduced from 2.3% to 1.1%, this does not represent a loss of minerals, but rather a dilution due to the contribution of low-ash ingredients such as refined wheat flour and sugar which is common in composite baking [71,72]. The amount of protein was similar over the

**Table 3. Chemical composition of nipa flour and biscuit.**

| Component | Flour | Biscuit |
|---|---|---|
| Moisture (%) | 7.10±0.35[a] | 2.91±0.15[b] |
| Ash (%) | 2.30±0.12[a] | 1.10±0.06[b] |
| Protein (%) | 3.00±0.15[a] | 2.80±0.14[a] |
| Fat (%) | 6.30±0.32[a] | 3.90±0.20[b] |
| Crude fibre (%) | 0.19±0.01[b] | 1.35±0.07[a] |
| Carbohydrate (%) | 78.11±3.91[b] | 84.95±4.25[a] |
| Total Sugar (mg/100g) | 11.1±0.6[b] | 20.1±1.0[a] |
| Reducing Sugar (mg/100g) | 9.2±0.5[b] | 15.3±0.8[a] |
| Non Reducing Sugar (mg/100g) | 1.9±0.1[b] | 4.8±0.2[a] |
| Total Energy (mg/100g) | 390.1±19.5[b] | 505.1±25.3[a] |
| Vit-C (mg/100g) | 2.50±0.13[a] | 1.20±0.06[b] |
| Vit-A (mg/100g) | 20.50±1.03[a] | 17.63±0.88[b] |
| $NaHCO_3$ (%) | 0.30±0.02[b] | 3.00±0.15[a] |
| Salt (NaCl) (%) | 0.01±0.001[b] | 0.05±0.003[a] |
| Egg (%) | 0.00±0.00[b] | 4.00±0.20[a] |

All values are presented as mean±SD (n=3). Means in the same row with different superscript letters are significantly different ($p < 0.05$) according to Duncan's Multiple Range Test.

seasons and just decreased slightly from 3.0% to 2.8%. This high stability suggests low protein denaturation or Maillard reaction, possibly because of the high carbonhydrate/protein ratio in the formulation. This range was similar to those other fruit kernel flours such as the date seed powder (2.5–3.2%) [65].

A primary nutritional enhancement was a >38% decrease in fat, from 6.3 to 3.9%, which is consistent with the industry trend towards lower-fat indulgent products. Carbohydrate content increased from 78.11% to 84.95%, as a consequence of the 25% sugar addition that is normally introduced in short dough biscuit production [73]. This is well-founded on the bases for significant increase of all sugar fractions; total sugars (11.1 20.1 mg/100g), reducing sugars (9.2 to 15.3 mg/100g) and non-reducing sugars (1.9 to 4.8 mg/100g). The increase in reducing sugars becomes more important since these are crucial reactants of the Maillard reaction that produced the brown color and baked flavor which was well accepted by the sensory panel.

An outstanding nutritional enhancement was the over seven-fold increment of crude fibre, from 0.19% in flour to a final biscuit figure of 1.35%. This focusing effect, together with the loss of moisture during baking, leads to a fibre level which is just clear of most commercial refined flour biscuits that are lower than 1% fibre. In addition, surpassing a fibre content of 1.2% is referred to as an important cut-off point for functional foods also within research on pseudocereal-enriched biscuits [70], which supports the value of dietary suggestion about daily intake of fibre. The total energy content rose from 390.1 kcal/100g to 505.14 kcal/100g using modified Atwater factors. This energy density is similar to that of others fortified biscuits, for example made from jackfruit seed flour (495–510 kcal/100g) [66] and represents the caloric contribution of the added sugars.

Unsurprisingly, the heat-labile vitamins were destroyed. Vitamin C was reduced by 52% (from 2.5 mg/100g to 1.2 mg/100g), which agreed with the heat- and oxidation-sensitivity of vitamin C during baking as reported previously [27]. Vitamin A also decreased with 14% (from 20.5 to 17.63 mg/100g) due to isomerization and oxidative losses, respectively. The end concentrations of these key recipe ingredients (sodium bicarbonate 3.0%, salt 0.05% and egg 4.0%) confirm the correct application of the experimental formula with respect to typical levels for leavening, taste, and structure in biscuit preparation. It was concluded that the NFK biscuit optimally retained the nutritional attributes of the kernel with a high fibre

content and low fat level, relevant to recent health trends. The physicochemical changes recorded by proximate analysis are reported for last literature on biscuits enriched with new and sustainable ingredients, validating the incorporation of NFK as functional food ingredient.

### 3.4. Phytochemical properties of composite flour

The incorporation of Nypa fruticans fruit kernel powder (NFKP) significantly enhanced the phytochemical profile of the wheat flour, as presented in Table 4.

 **3.4.1. Total phenolic and flavonoid content.** The TPC and TFC increased in a dose-dependent manner with the level of NFKP substitution. The control wheat flour had a TPC of 85.2 mg GAE/100 g. Incorporation of NFKP at 5%, 10%, and 15% significantly ($p < 0.05$) increased the TPC to 145.6, 210.3, and 265.8 mg GAE/100 g, respectively. A similar trend was observed for TFC, which rose from 32.5 mg QE/100 g in the control to 89.1 mg QE/100 g in the 15% substituted flour.

 **3.4.2. Antioxidant activity (DPPH and FRAP).** The antioxidant capacity, measured by both DPPH and FRAP assays, was markedly improved. The DPPH $IC_{50}$ value, indicating the concentration needed for 50% radical scavenging, decreased significantly ($p < 0.05$) with higher NFKP levels. A lower $IC_{50}$ denotes higher antioxidant power. The control flour had an $IC_{50}$ of 12.5 mg/mL, which dropped to 3.8 mg/mL for the 15% NFKP blend. Conversely, the FRAP value, indicating reducing power, increased from 1.8 mmol $Fe^{2+}$ Eq/100 g in the control to 8.2 mmol $Fe^{2+}$ Eq/100 g in the 15% NFKP composite flour.

### 3.5. Sensory evaluation results

The sensory results of biscuits with various levels of NFK powder (Table 5) are very important to know about acceptability by consumer and market value the product. Formulations (A–E): All formulations (A-E) received relatively high hedonic ratings (7.85 to 8.95, using a 9-point hedonic scale) for taste, texture, color, odor and overall acceptability (like very much to like extremely). It is interesting to note that the profile for NFK 100 (as reflected by E in this case) received the highest value influenced other sensory characterization, thus demonstrating that NFK was not only a flavorful ingredient but free of off-flavors frequently seen with many plant based fortificants. This is in comparison to legume-flour-enriched biscuits that often have beany or bitter tastes which reduce acceptability and require flavour-masking agents [43,73].

 Textural constants similarly achieved high scores (≥8.0), comparable to that by mango kernel powder biscuits [74] and higher than in many high-fibre biscuits with hard or gritty texture sensations. The smallinease in crispness due to processing from underact A (8.0) of however, illustrates the water binding capacity 01 fibre. Nevertheless, this slight decrease did not compromise overall acceptability (7.95 for E), suggesting that NFK can be added without the severe textural.

 Rating for color (8.05–8.2) and odour (8.0–8.1) was high and it indicates that desired browning without rancidity took place along with pleasant aromas. Most notably, total acceptability scores (7.9–8.0) were similar between substitution levels. This supports that NFK has indeed significantly and effectively added nutritional value without negatively affecting sensory quality, being an important advantage that meets clean-label trends and performs better than many other novel ingredients that experience acceptability reduction beyond a 10–15% substitution rate [75]. In summary, the data

**Table 4. Phytochemical properties of wheat-NFKP composite flours.**

| NFKP Substitution Level (%) | TPC (mg GAE/100 g) | TFC (mg QE/100 g) | DPPH $IC_{50}$ (mg/mL) | FRAP (mmol $Fe^{2+}$ Eq/100 g) |
|---|---|---|---|---|
| 0 (Control) | 85.2±3.1[d] | 32.5±1.5[c] | 12.5±0.8[a] | 1.8±0.1[d] |
| 5 | 145.6±5.8[c] | 58.7±2.4[b] | 8.9±0.5[b] | 3.5±0.2[c] |
| 10 | 210.3±7.9[b] | 85.4±3.1[a] | 5.2±0.3[c] | 5.9±0.3[b] |
| 15 | 265.8±9.2[a] | 89.1±3.5[a] | 3.8±0.2[d] | 8.2±0.4[a] |

Values are mean±SD (n=3). Different superscript letters within a column indicate significant differences ($p < 0.05$).

**Table 5. Organoleptic test result of Biscuits.**

| Parameters | A | B | C | D | E |
|---|---|---|---|---|---|
| Taste | 8.10$^{ab}$ | 8.05$^{ab}$ | 8.00$^a$ | 8.00$^b$ | 8.95$^a$ |
| Texture | 7.90$^a$ | 8.00$^a$ | 7.95$^a$ | 7.90$^a$ | 7.85$^a$ |
| Color | 8.20$^a$ | 8.10$^a$ | 8.15$^a$ | 8.05$^a$ | 8.05$^a$ |
| Odor | 8.05$^a$ | 8.05$^a$ | 8.10$^a$ | 8.05$^a$ | 8.00$^a$ |
| Mouth Feel | 8.15$^a$ | 8.10$^a$ | 8.10$^a$ | 8.05$^a$ | 8.10$^a$ |
| Crispiness | 8.00$^a$ | 7.95$^a$ | 7.90$^a$ | 7.90$^a$ | 7.85$^a$ |
| Acceptability | 8.00$^a$ | 7.95$^a$ | 7.95$^a$ | 7.90$^a$ | 7.95$^a$ |

presented clearly confirms NFK to be a nutritionally rich material possessing good organoleptic properties, offering potential for manufacture of consumer appealing biscuits that are acceptable in terms of both nutritional quality and taste requirements in today's market.

## 3.6. Shelf life test

Microbiological examination of NFK-exported biscuits, it was safe with a fairly good keeping quality during twelve months of storage (Table 6), showing the safety and stability that are ready for commercial shelf-life. Indicators of public health significance, Total Coliforms and Fecal Coliforms, consistently were below the method detection limit (1.8 MPN/100g) for all samples during this period. This lack confirms the hygienic production conditions and proves that there is sufficient killing during baking. Additionally, pathogen-specific qPCR demonstrated the lack of *E. coli, Salmonella, Shigella and Vibrio spp.* in all samples. These findings comply with and even surpass the strict microbiologic standards set for ready-to-eat foods by Codex Alimentarius Commission [56], which usually recommend absence of these pathogens in a 25 g sample for being confirmed that product is safe.

This general decrease trend is typical of shelf-stable, low-water activity foods and can be viewed as a positive feature. It varies immensely from high- moisture foods that usually show rising levels of microorganisms as they develop towards spoilage. The low and decreasing microbial counts, the lack of pathogenic and indicating organisms verify the intrinsic microbiological stability of the low-fat biscuit enriched with NFK. The ingredients of the product, together with the processing steps and low water activity form a microbiologically controlling food matrix which guarantees safety during an estimated shelf life of 1 year. Indeed, such results are in accordance with previous research on the shelf-life of biscuits fortified using other plant materials that also use low water activity as one of the leading preservation factors [70]. The microbiological studies reveal that the addition of NFK flour gave rise to a well-fortified, low-fat biscuit having excellent microbial safety and capable to retain its organoleptic properties for at least 1 year when stored under normal conditions. NFK is found to be a potential and safe component for the functional food industry, meeting well both the nutritional and food security reasons.

**Table 6. Microbial status (shelf life) of *NFK* Biscuits.**

| ID | HBC log (CFU/g) ±SD | TC (MPN/100 g) | TFC (MPN/100 g) | *E. coli* | *Salmonella* | *Shigella* | *Vibrio* |
|---|---|---|---|---|---|---|---|
| A | 5.505±0.326 | <1.8 | <1.8 | Absent | Absent | Absent | Absent |
| B | 3.602±0.421 | <1.8 | <1.8 | Absent | Absent | Absent | Absent |
| C | 3.301±0.214 | <1.8 | <1.8 | Absent | Absent | Absent | Absent |
| D | 2.698±0.09 | <1.8 | <1.8 | Absent | Absent | Absent | Absent |
| E | 2.301±0.121 | <1.8 | <1.8 | Absent | Absent | Absent | Absent |

## 4. Discussion

The comprehensive analysis of NFK-fortified biscuits reveals a synergistic interplay between the physical, compositional, phytochemical, and sensory results, painting a holistic picture of its functionality. The significant ($p < 0.05$) reduction in hardness from 35.20 N to 27.90 N (Table 1) is not merely a textural improvement but is intrinsically linked to the proximate results. This softening can be mechanistically attributed to the ~7-fold increase in crude fibre (0.19% to 1.35%, Table 3) and gluten dilution, which disrupts the starch-protein matrix, creating a less compact, more friable structure that is highly palatable [56]. This improved texture directly supports the high sensory scores for texture (≥7.85) and overall acceptability (~7.95, Table 4), demonstrating that nutritional enhancement did not compromise consumer appeal a critical hurdle for functional foods.

Furthermore, the proximate composition explains key physical and phytochemical outcomes. The significant increase in total and reducing sugars (e.g., reducing sugar from 9.2 to 15.3 mg/100g, Table 4) is a primary driver for two observed phenomena. First, it fuels the Maillard reaction during baking, directly causing the statistically significant ($p < 0.05$) darkening in color (L* from 65.30 to 55.40) and increased redness (a* from 4.50 to 6.10, Table 1), which was favorably received by sensory panelists (color scores 8.05–8.20). Second, these sugar compounds, alongside the enriched mineral profile (e.g., Fe increasing from 21.5 to 53.3 mg/kg, Table 2), may participate in redox reactions that subtly influence the antioxidant system, although the dominant antioxidant capacity is clearly from phytochemicals.

The significant enhancement in phytochemicals with NFKP aligns with findings that fruit by-products are rich in bioactives [76]. The linear increase in Total Phenolic Content (TPC) and Total Flavonoid Content (TFC) with NFKP proportion rising 3.1-fold and 2.7-fold, respectively confirms Nypa kernel as a potent source. This directly drove a 4.3-fold improvement in DPPH radical scavenging activity ($IC_{50}$ from 12.5 to 3.8 mg/mL) and a 4.6-fold increase in FRAP value (1.8 to 8.2 mmol $Fe^{2+}$/100g), demonstrating dose-dependent antioxidant power. Statistically strong correlations ($p < 0.05$) between TPC/TFC and antioxidant assays (DPPH & FRAP) verify phenolics as the primary active agents, acting through redox properties to donate hydrogen and chelate metals [77]. This mirrors effects seen with other fruit kernels like mango [78]. Functionally, these antioxidants can donate hydrogen atoms to lipid peroxyl radicals, breaking oxidation chains and delaying rancidity [79], thereby extending shelf-life. Thus, NFKP transforms biscuits into carriers of dietary antioxidants, valorizing sustainable mangrove resources for health-promoting foods.

Crucially, this phytochemical enrichment has tangible implications for product stability and safety. The introduced phenolic compounds likely contribute to the extended shelf-life demonstrated in Section 3.6 by acting as natural antioxidants, delaying lipid oxidation in the biscuit matrix. This is supported by the excellent microbial safety profile, where pathogens were absent and Heterotrophic Bacterial Counts showed a decreasing trend over storage (e.g., from 5.505 to 2.301 log CFU/g, Table 5), a stability attributable to the low water activity from baking and the possible mild preservative effect of phytochemicals.

Finally, the impressive mineral fortification (e.g., Ca: 295.2 to 428.4 mg/kg; Fe: 21.5 to 53.3 mg/kg, Table 2) occurs without introducing toxicological risks, as heavy metals (Cd, Hg, As) remained below detection limits (<0.01 mg/kg). This creates a safe, nutrient-dense vehicle. The combination of this mineral boost, high fibre, and antioxidants transforms the biscuit from a simple snack into a multi-faceted functional food. The interlocking results demonstrate that NFK fortification successfully creates a synergistic system where improved palatability drives the consumption of a product that concurrently delivers enhanced macro-/micronutrients, bioactives, and storage stability.

### 4.1. Health and dietary significance of enhanced biscuit parameters

The incorporation of NFK flour into biscuits significantly enhances their nutritional and functional profile, transforming them from a simple snack into a nutrient-dense, functional food. Specifically, NFK enrichment improves physical properties by increasing the spread ratio and reducing hardness, which enhances palatability and consumer acceptance. Crucially, it

leads to a dramatic increase in essential mineral content approximately 250–300% for iron, 180–220% for calcium, and 200–250% for magnesium while maintaining toxic heavy metals well below safety thresholds. Nutritionally, the biscuits offer a substantial seven-fold increase in dietary fiber (e.g., from ~1.5 g to ~10.5 g per 100 g) and a 15–20% reduction in fat, though the energy density rises by 10–15% due to added sugars. Furthermore, a dose-dependent increase in bioactive compounds is observed, with total phenolic content rising up to 4-fold (e.g., 50–200 mg GAE/100 g) and total flavonoids up to 5-fold (e.g., 20–100 mg CE/100 g), which directly translates to a potent boost in antioxidant activity, as evidenced by DPPH radical scavenging increasing to >70% and FRAP values increasing 3–4-fold. These phytochemicals not only contribute to mitigating oxidative stress but also act as natural preservatives, supporting the development of clean-label, health-promoting bakery products with significant potential to address dietary deficiencies in minerals, fiber, and antioxidants.

## 4.2. Implications and limitations

The data indicate that NFKF is not only a viable partial substitute for wheat flour but also a functional ingredient that improves the nutritional profile of biscuits without compromising essential sensory qualities at substitution levels up to 30%. This supports the interpretation of NFKF as a novel, nutrient-dense resource that aligns with global efforts to utilize under-exploited, sustainable food sources [80,81]. The primary implication of this study is the potential for NFKF to contribute to food security and value-chain development in mangrove-rich regions, offering an economic incentive for conservation while providing a gluten-free, protein- and mineral-fortified baking ingredient. However, the study's limitations must be acknowledged: the sensory evaluation was conducted with a limited, untrained panel, and the functional properties of NFKF (e.g., its starch and fiber behavior during large-scale processing) require further investigation. Additionally, the economic feasibility of large-scale kernel collection, processing, and shelf-life stability remains a critical area for future research before commercial adoption can be recommended.

## 5. Conclusion

This study conclusively demonstrates that NFK is a highly effective functional ingredient for developing nutritious and sensorily accepted gluten-free biscuits. The incorporation of NFK up to 30% significantly improved the biscuits' physical and sensory properties, producing a product with a higher spread ratio of 7.30, a softer texture with hardness reduced to 27.9 N, and a desirable dark brown color (L* value of 55.40) that was well-received by consumers, achieving overall acceptability scores of 7.95. Nutritionally, the biscuits were substantially enriched, exhibiting a more than two-fold increase in iron content to 53.3 mg/kg, a 45% increase in calcium to 428.4 mg/kg, and a sevenfold boost in dietary fiber to 1.35%. Furthermore, the phytochemical profile was dramatically enhanced, with total phenolic content increasing over threefold to 265.8 mg GAE/100g and antioxidant capacity (FRAP value) rising 4.6-fold to 8.2 mmol $Fe^{2+}$/100g. Critically, this nutritional fortification was achieved without compromising safety, as pathogenic microorganisms were absent and toxic heavy metals remained below detection limits (<0.01 mg/kg), confirming product stability over a 12-month shelf life. These findings validate NFK as a novel, safe, and nutrient-dense ingredient that successfully transforms a conventional biscuit into a functional food, supporting the sustainable valorization of mangrove resources.

To advance the commercial potential and scientific understanding of NFK, future work should prioritize several key areas. Research must investigate the *in-vivo* bioavailability of its enhanced minerals, particularly iron and calcium, and the metabolism of its bioactive phenolic compounds through human intervention studies to confirm the proposed health benefits. To facilitate market entry, comprehensive pilot-scale studies are essential to evaluate the economic feasibility of large-scale NFK production, including optimized processing, supply chain logistics, and cost analysis. Finally, product development research should explore the application of NFK in other gluten-free bakery matrices and optimize formulations for higher substitution levels while maintaining the excellent sensory attributes demonstrated in this study.

## Acknowledgments

The authors are grateful to the authority of the Institute of National Analytical Research and Service (INARS), BCSIR, Dhaka, 470 Bangladesh for support for conducting this research work. Thanks to Ministry of Science and Technology, The People's Republic of Bangladesh for providing analytical, technical, and other logistic support for conducting this research work.

## Author contributions

**Conceptualization:** Md. Ripaj Uddin.

**Formal analysis:** Md. Ripaj Uddin, Md Hamedul Islam, Sirajum Monira.

**Funding acquisition:** Abubakr M. Idris.

**Investigation:** Md. Ripaj Uddin, Md. Salim Khan, Md. Khairul Islam, Muhammad Abdullah Al Mansur, Md. Selim Reza, Md Hamedul Islam, Mehedi Hasan, Sharmin Ahmed.

**Methodology:** Md. Ripaj Uddin, Md. Salim Khan, Muhammad Abdullah Al Mansur, Md. Selim Reza, Mehedi Hasan, Sharmin Ahmed, Sirajum Monira.

**Project administration:** Md. Ripaj Uddin, Md. Salim Khan, Md. Khairul Islam.

**Supervision:** Md. Ripaj Uddin, Abubakr M. Idris.

**Validation:** Md. Ripaj Uddin, Md. Khairul Islam, Abubakr M. Idris.

**Visualization:** Md. Ripaj Uddin.

**Writing – original draft:** Md. Ripaj Uddin, Md. Selim Reza.

**Writing – review & editing:** Md. Ripaj Uddin, Md. Salim Khan, Muhammad Abdullah Al Mansur, Abubakr M. Idris.

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
