## [Decision Letter · Decision Letter 0]

7 Dec 2025

Dear Dr. Md. Ripaj Uddin,

Thank you for submitting your manuscript to PLOS ONE. After careful consideration, we feel that it has merit but does not fully meet PLOS ONE’s publication criteria as it currently stands. Therefore, we invite you to submit a revised version of the manuscript that addresses the points raised during the review process.

We look forward to receiving your revised manuscript.

Kind regards,

Muhammad Sibt-e-Abbas

Academic Editor

PLOS ONE

Journal Requirements:

4. Please be informed that funding information should not appear in the Acknowledgments section or other areas of your manuscript. We will only publish funding information present in the Funding Statement section of the online submission form. Please remove any funding-related text from the manuscript.

Additional Editor Comments (if provided):

The article is well structured but still has many grey areas that need to be addressed and the manuscript must be thoroughly improved before consideration for publication.

Reviewers' comments:

Reviewer's Responses to Questions

**Comments to the Author**

1. Is the manuscript technically sound, and do the data support the conclusions?

Reviewer #1: Partly

Reviewer #2: Partly

2. Has the statistical analysis been performed appropriately and rigorously?

Reviewer #1: Yes

Reviewer #2: No

3. Have the authors made all data underlying the findings in their manuscript fully available?

Reviewer #1: Yes

Reviewer #2: Yes

4. Is the manuscript presented in an intelligible fashion and written in standard English?

Reviewer #1: Yes

Reviewer #2: Yes

Reviewer #1: Nypa fruticans (Mangrove Palm) Fruit Kernel: A Novel Ingredient for Nutrient-Dense, Low-Fat Biscuits

Manuscript ID: PONE-D-25-57201

Review 1

17 November 2025

Dear Editor of Plos One

The manuscript has to be improved. The Introduction section has to be better treated. M&M section has to be detailed. The R&D section has to be better discussed. The bibliography has to be improved. Inaccuracies in the manuscript. The manuscript has to be organised as per the instructions to authors.

I suggest a major revision

To Authors (in detail):

1) The argument is interesting but it has to be improved. The Introduction section has to be better treated. M&M section has to be detailed. The R&D section has to be better discussed. The bibliography has to be improved. Inaccuracies in the manuscript. The manuscript has to be organised as per the instructions to authors.

2) Apply carefully the instructions to authors in the whole manuscript and use some recent published paper as a sample;

3) In the title and at the first mention in the manuscript, when you write a scientific name, apply the International Botanical nomenclature and, after the species, insert the abbreviation of the Botanist (not italicised);

4) Please, include the page number;

5) Abstract section, insert some relevant numeric result;

6) Keywords section, write the keywords in alphabetical order;

7) Verify if to number the sections and the sub-sections;

8) Introduction section, lines 46-52, explain that studies were conducted to reduce the shortening by replacement of butter with a vegetable oil. Support with proper reference [X1]:

[X1] Effects of shortening replacement with extra virgin olive oil on the physical–chemical–sensory properties of Italian Cantuccini biscuits. Foods 2022, 11, 299

https://doi.org/10.3390/foods11030299

9) Introduction section, lines 48-49, this is a very important statement but poorly supported with proper references. Find, read and discuss [X2]:

[X2] Techno-functional, antioxidants, microstructural, and sensory characteristics of biscuits as affected by fat replacer using roasted and germinated chickpea (Cicer arietinum L.).

International Journal of Food Properties, 26:1, 2055-2077 (2023).

DOI: 10.1080/10942912.2023.2242602

10) Introduction section line 71 and in the whole manuscript, when you write the references numbers between brackets, separate them by a comma instead of a semicolon;

11) Introduction section line 72 and in the whole manuscript, when you write a weight, separate the numeric value and the unit: 100 g instead of 100g;

12) Sub sub-section 2.1.1., replace 02 January with 2 January;

13) Caption of figures: “Fig 1.” instead of “Fig.1.”;

14) Sub sub-section, detail the degree of ripeness of Nypa fruticans fruits. Do not write “Full ripeness” or “Industrial Fruit ripeness”. Use days after blossoming or some other recognizable degree of ripeness;

15) Sub-section 2.2, line 122 and in the whole manuscript, tables and figures: when you write a temperature, separate the numeric value and the unit: 105 °C instead of 105°C;

16) Sub-section 2.6, and in the whole manuscript, when you write a date, apply the English form to differentiate the day of the month and the month. If you want, you can write the abbreviation of the month: Jan instead of 01;

17) Line 327 and in the whole manuscript, tables and figures: 24 h instead of 24h, separate;

18) Caption of tables, after the table number, insert a point, use some recently published paper as a sample;

19) Line 357 and in the whole manuscript, when you write a length separate the numeric value and the unit: 55.15 mm instead of 55.15mm;

20) Table 2 and in the whole manuscript, kg always in small letters;

21) Sub-section 3.3., line 449 and in the whole manuscript, the abbreviation NFK in capital letters;

22) Line 531, when you write a weight, separate the numeric value and the unit: 25 g instead of 25g, separate;

23) Table 5, the International abbreviation for grams is g and not gm;

24) Caption of table 5, after the table number, replace the colons with a point;

25) Discussion section, discuss about the interaction between the results of your different analyses;

26) Discussion section, extend the discussion of each parameter in relation to the human diet;

27) The discussion has to be supported by your statistical data;

28) Before the conclusion section insert a brief section discussing interpretation, implications and limitations of your study;

29) References section apply the instructions to authors and use some recently published paper as a sample;

30) Write in red color the corrections you will do.

I suggest a major revision

Regards.

Reviewer #2: The study focuses on Nypa fruticans (Mangrove Palm) Fruit Kernel: A Novel Ingredient for Nutrient-Dense,

Low-Fat Biscuits. This study is very relevant and important since there is a need to use under fruits by products in the development of new food products. This is important since these fruit by products are rich in health promoting compounds. I have highlighted my comments and suggestions in the main document and the authors must address all of them, point by point. Below is the summary of the report:

(1) Title, it needs to be revised to be in line with the content of the manuscript, i.e. Nypa fruticans (Mangrove Palm) Fruit Kernel: Effect on physicochemical, sensory evaluation and shelf-life of wheat biscuits.

(2) Abstract, should be straight to the point since it is a summary of results. Under color, the results of L*, a* and b* values should be clear rather saying darker colour.

(3) Introduction, it is sound, however, the knowledge gap and the novelty of the study is lacking. What is the current problem with wheat biscuits? How is the inclusion of Nypa fruticans fruit kernel addressing the problem? Previous studies done on the subject matter should be included to show what is lacking in the literature.

(4) Materials and methods, it is very sound, the procedures for all the methods were explained in detail. However, section 2.2 should be summarized since the methods are known (AOAC standard methods).

(5) Results and discussion, this section is very weak and must be strengthened to improve the scientific tone. There is a need to do statistical analysis for most of the tables. Otherwise, the discussion is meaningless if the data on the tables is speculated. The authors compared the data of raw flour and biscuit, but they did not indicate the sample used and why they chose the sample. This is important, authors must not speculate the biscuit sample, the information should be included under materials and methods. For color analyis, the discussion should be expanded by explaining the reason for increase in a* and decrease in b* values. The sensory evaluation section must be revised by doing statistical analysis of the results. The reason(s) should be explained in detail for sample with the highest value for each sensory attribute.

(6) Conclusion, should be in line with the objective of the study and should include practical implication of the obtained data.

(7) References, I did not page numbers, but the style is consistent.

(8) General comment, fruits by products are rich in phytochemicals, thus adding this section (total phenolic, total flavonoid, DPPH and FRAP) will improve the quality of the manuscript.

**Do you want your identity to be public for this peer review?** For information about this choice, including consent withdrawal, please see our Privacy Policy

Reviewer #1: No

Reviewer #2: No

---

## [Author Response · Author response to Decision Letter 1]

26 Dec 2025

Q1. Please ensure that your manuscript meets PLOS ONE's style requirements, including those for file naming. The PLOS ONE style templates can be found at

Response: The manuscript has been formatted according to the PLOS ONE template.

Q2. In your Methods section, please provide additional information regarding the permits you obtained for the work. Please ensure you have included the full name of the authority that approved the field site access and, if no permits were required, a brief statement explaining why.

Response: No special permission was required to access the field site, as local laws do not restrict access to the area. Samples were collected from the coastal zone, not from deep forest.

Q3. We note that the grant information you provided in the ‘Funding Information’ and ‘Financial Disclosure’ sections do not match.

Response: We have provided the correct grant numbers in the 'Funding Information' section.

Q4. Please be informed that funding information should not appear in the Acknowledgments section or other areas of your manuscript. We will only publish funding information present in the Funding Statement section of the online submission form. Please remove any funding-related text from the manuscript.

Response: The funding information has been removed from the manuscript.

Q5. If the reviewer comments include a recommendation to cite specific previously published works, please review and evaluate these publications to determine whether they are relevant and should be cited. There is no requirement to cite these works unless the editor has indicated otherwise.

Response: We have reviewed the recommended journals and have cited only those relevant to our work.

---

## [Editor Report · Decision Letter 1]

22 Jan 2026

Dear Dr. Uddin,

We look forward to receiving your revised manuscript.

Kind regards,

Muhammad Sibt-e-Abbas

Academic Editor

PLOS One

**Journal Requirements:**

**Additional Editor Comments:**

Reviewer 1

Comment 3

In the title and at the first mention in the manuscript, when you write a scientific name, apply the International Botanical nomenclature and, after the species, insert the abbreviation of the Botanist (not italicised);

Response: The title has been corrected in accordance with your suggestion.

Academic Editor: Not satisfied.

Reviewer 1

Comment 7

Verify if to number the sections and the sub-sections;

Response: We have corrected the relevant section and subsection.

Academic Editor: Section 2.6 and section 4 appear twice.

Please make these 2 changes and resubmit for final decision of your manuscript

---

## [Author Response · Author response to Decision Letter 2]

26 Jan 2026

Query: In your Methods section, please provide additional information regarding the permits you obtained for the work. Please ensure you have included the full name of the authority that approved the field site access and, if no permits were required, a brief statement explaining why.

Response: This addition has been incorporated into the revised manuscript within the Methodology section (lines 116-122).

Fresh fruits of Nypa fruticans were collected from the Sundarbans Mangrove Forest or specific river estuary in Bangladesh. No specific permits were required from Forest Department of Bangladesh for the collection of Nypa fruticans fruits at this site, as the area is publicly accessible and the species is not protected or endangered under national or local regulations in this region. The collection was for academic research purposes and did not involve disturbance of protected land or threatened wildlife.

---

## [Editor Report · Decision Letter 2]

27 Jan 2026

Nypa fruticans Wurmb (Mangrove Palm) Fruit Kernel: Effect on physicochemical, sensory evaluation and shelf-life of wheat biscuits

PONE-D-25-57201R2

Dear Dr. Md. Ripaj Uddin,

We’re pleased to inform you that your manuscript has been judged scientifically suitable for publication and will be formally accepted for publication once it meets all outstanding technical requirements.

Kind regards,

Muhammad Sibt-e-Abbas

Academic Editor

PLOS One

Additional Editor Comments (optional):

Thank you for your compliance with the review process. I am satisfied with the changes made in the manuscript as per the suggestions of reviewers.
---

## [Editor Report · Acceptance letter]

PONE-D-25-57201R2

PLOS One

Dear Dr. Uddin,

I'm pleased to inform you that your manuscript has been deemed suitable for publication in PLOS One. Congratulations! Your manuscript is now being handed over to our production team.

Kind regards,

on behalf of

Dr. Muhammad Sibt-e-Abbas

Academic Editor

PLOS One